# Seroprevalence and longevity of SARS-CoV-2 nucleocapsid antigen-IgG among health care workers in a large COVID-19 public hospital in Saudi Arabia: A prospective cohort study

Faisal Alasmari[1,2]*, Mahmoud Mukahal[1], Alaa Ashraf Alqurashi[3], Molla Huq[4], Fatima Alabdrabalnabi[1], Abdullah AlJurayyan[5], Shymaa Moshobab Alkahtani[6], Fatimah Salem Assari[6], Rahaf Bashaweeh[6], Rana Salam[7], Solaf Aldera[1], Ohud Mohammed Alkinani[8], Talal Almutairi[9], Kholoud AlEnizi[10], Imad Tleyjeh[2]

1 Infection Control and Environmental Health Administration, King Fahad Medical City, Riyadh, Saudi Arabia, 2 Infectious Diseases Section, King Fahad Medical City, Riyadh, Saudi Arabia; College of Medicine, Al Faisal University, Riyadh, Saudi Arabia, 3 Department of Epidemiology and Preventive Medicine, Monash University, Melbourne, Australia, 4 Immunology and Serology Laboratory, King Fahad Medical City, Riyadh, Saudi Arabia, 5 College of Medicine, Al Faisal University, Riyadh, Saudi Arabia, 6 Public Health College, Saudi Electronic University, Riyadh, Saudi Arabia, 7 Infectious Diseases Section, King Fahad Medical City, Riyadh, Saudi Arabia, 8 Pathology and Clinical Laboratory Administration, King Fahad Medical City, Riyadh, Saudi Arabia, 9 Radiology Service Administration, King Fahad Medical City, Riyadh, Saudi Arabia, 10 Research Center, King Fahad Medical City, Riyadh, Saudi Arabia

* faalasmari@kfmc.med.sa

## Abstract

Seroprevalence of SARS-CoV-2 IgG among health care workers (HCWs) is crucial to inform infection control programs. Conflicting reports have emerged on the longevity of SARS-CoV-2 IgG. Our objective is to describe the prevalence of SARS-CoV-2 IgG in HCWs and perform 8 months longitudinal follow-up (FU) to assess the duration of detectable IgG. In addition, we aim to explore the risk factors associated with positive SARS-CoV-2 IgG. The study was conducted at a large COVID-19 public hospital in Riyadh, Saudi Arabia. All HCWs were recruited by social media platform. The SARS-CoV-2 IgG assay against SARS-CoV-2 nucleocapsid antigen was used. Multivariable logistic regression was used to examine association between IgG seropositive status and clinical and epidemiological factors. A total of 2528 (33% of the 7737 eligible HCWs) participated in the survey and 2523 underwent baseline serological testing in June 2020. The largest occupation groups sampled were nurses [n = 1351(18%)], physicians [n = 456 (6%)], administrators [n = 277 (3.6%)], allied HCWs [n = 205(3%)], pharmacists [n = 95(1.2%)], respiratory therapists [n = 40 (0.5%)], infection control staff [n = 21(0.27%), and others [n = 83 (1%)]. The total cohort median age was 36 (31–43) years and 66.3% were females. 273 were IgG seropositive at baseline with a seroprevalence of 10.8% 95% CI (9.6%-12.1%). 165/185 and 44/112 were persistently IgG positive, at 2–3 months and 6 months FU respectively. The median (25th–75th percentile) IgG level at the 3 different time points was 5.86 (3.57–7.04), 3.91 (2.46–5.38), 2.52 (1.80–3.99) respectively. Respiratory therapists OR 2.38, ($P$ = 0.035), and those with hypertension OR = 1.86, ($P$ = 0.009) were more likely to be seropositive. A high

**Data Availability Statement:** The data underlying the results presented in the study are available from (DOI ).

**Funding:** The author(s) received no specific funding for this work.

**Competing interests:** The authors have declared that no competing interests exist.

proportion of seropositive staff had prior symptoms 214/273(78%), prior anosmia was associated with the presence of antibodies, with an odds ratio of 9.25 (P<0.001), as well as fever and cough. Being a non-smoker, non-Saudi, and previously diagnosed with COVID-19 infection by PCR were statistically significantly different by seroprevalence status. We found that the seroprevalence of IgG against SARS-CoV-2 nucleocapsid antigen was 10.8% in HCWs at the peak of the pandemic in Saudi Arabia. We also observed a decreasing temporal trend of IgG seropositivity over 8 months follow up period.

## Introduction

Severe acute respiratory syndrome coronavirus 2 (SARS-CoV-2) has caused a global pandemic. The first case of SARS-CoV-2 infection in Saudi Arabia was reported in Eastern province January 20,2020, with 545829 infections and 8610 deaths as of 11 September 2021 [1].

Health care workers (HCWs) are at an increased risk of becoming infected with SARS-CoV-2. Understanding the prevalence of SARS-CoV-2 carriage amongst HCWs is crucial to help monitor transmission dynamics and inform the development of screening programs. Access to COVID-19 molecular testing during early time of pandemic was mainly confined to symptomatic individuals, and therefore the rates of infection in asymptomatic or minimally symptomatic HCWs have been difficult to determine.

Serological testing can detect prior SARS-CoV-2 infection for which nasopharyngeal sampling resulted in false negatives or for which reverse transcription-PCR testing was not performed. It requires high sensitivity and specificity, especially when seroprevalence is low, in order to have an acceptable positive predictive value [2].

Several seroprevalence studies of HCWs from different countries in the first phase of the pandemic revealed a wide range of seropositivity [3–6]. The reasons for such variation may reflect the underlying community transmission rate in addition to an increased risk in certain hospital.

Conflicting reports have been published on the longevity of SARS-CoV-2 antibodies. For instance, an Iceland study showed IgG antibody levels to Nucleocapsid (NC) and the S1 component of spike were relatively stable in 1215 individuals for 100–125 days [7]. Similarly, data from 121 individuals suggest that IgG response to trimerized spike were sustained for 110 days post symptoms onset [8]. However, other reports have observed declines in IgG antibody levels over similar time periods [9–11].

Few studies have looked at HCWs across a healthcare system that included individuals with both direct patient care and non-clinical functions. In this study, we invited all HCWs to participate in a serologic survey from June 2020 to February 2021, after the first wave of SARS-CoV-2 infection which occurred from March through May 2020. We aimed to investigate the seroprevalence and its predictors and evaluate temporal trends in the levels of IgG antibody over an 8-month follow up period.

## Methods

### Setting and participants

King Fahad Medical City (KFMC) is a large referral facility that has a total of 7737 HCWs: 1021 medical staff, 3004 nursing staff, 1906 allied health personnel, and 1806 administrative personnel. It was assigned by the Ministry of Health as a public COVID-19 center.

All HCWs who could provide written informed consent were deemed eligible to be included in this study. There were no exclusion criteria except for actively symptomatic employees. HCWs were recruited via social media directed to the entire employee workforce. There was no predefined sample size. Participants self-reported for enrolment. Staff were asked to complete a survey on sociodemographic and clinical characteristics, job duties and location, COVID-19 symptoms, a self-reported polymerase chain reaction (PCR) test history with test date if available, travel record since January 2020, and exposure risks (patient, coworker, and household contact).

We planned to follow the IgG positive cohort over 1 year at regular time intervals. First time point was on June-July, 2020, and then on September, 2020 and finally on January-February 2021 to have approximately three months' interval between each test. This study was approved by the local institutional review board IRB log # 20–382, and written informed consent was obtained from all participants.

## Laboratory methods

Serum IgG to SARS-CoV-2 NC was measured using chemiluminescent microparticle immunoassay performed on an automated high throughput chemistry immunoanalyzer on the ARCHITECT i System. The resulting reaction is measured as a relative light unit (RLU). There is a direct relationship between the amount of IgG antibodies and the RLU.

Results are reported in RLU index; a value greater than or equal to 1.4 RLU is considered a positive antibody response. Values of more than 1.4 and < 3.99 were categorized as low values, and > 3.99 as high values. Though not a direct titer, higher index values highly correlate to neutralization titers [12].

The reported assay sensitivity is 100% with a specificity of 99% at greater than 14 days' post symptom onset, and at 5% prevalence, the positive predictive value is 93.4% and negative predictive value 100% [13].

## Statistical methods

Descriptive statistics (mean ± SD or Median (25th– 75th percentiles) were used to summarize patient characteristics. Bivariate testing was carried out using appropriate statistical tools based on variable type and distribution (e.g., chi-square test, Fisher's exact test, Wilcoxon's rank sum test and univariate linear or logistic regression analysis). Multivariable logistic regression was used to find association between serology positivity and different patient characteristics and COVID-19 symptoms. A p value of less than or equal to 0.05 was set as the threshold for statistical significance. All statistical analyses were performed using Stata 16.1 (Stata Corp, Texas).

## Results

Of the 7737 eligible HCWs, 2528 (33%) participated in the survey and 2523 underwent baseline serological testing. The largest occupation groups sampled were nurses [n = 1351(18%)], physicians [n = 456 (6%)], administrators [n = 277 (3.6%)], allied HCWs [n = 205(3%)], pharmacists [n = 95(1.2%)], respiratory therapists [n = 40(0.5%)], infection control staff [n = 21 (0.27%], and others [n = 83 (1%)]. The median (25th– 75th percentile) age was 36 years (31–43), and 66% were female. The other sociodemographic and clinical characteristics of the cohort are shown in Table 1 and Fig 1.

Overall, 273 HCWs had detectable IgG antibodies for SARS-CoV-2 with seroprevalence rate of 10.8% 95% CI (9.6%-12.1%). We observed highest prevalence rate among respiratory

**Table 1. Demographics and clinical characteristics of study participants.**

| Characteristics | All participants, n (%) | Seropositive, n (%) | Seronegative, n (%) | Seroprevalence, (%) | P value |
|---|---|---|---|---|---|
| N | 2528 | 273 (10.80%) | 2250 (89.00%) | 10.8% | |
| Age | 36 (19–71) | 37 (24–63) | 36 (19–71) | | 0.073 |
| Serology test 1 | 0.03 (0.02–0.07) | 5.86 (3.57–7.04) | 0.03 (0.02–0.05) | | |
| Serology test 2 | 3.73 (2.22–5.24) | 3.91 (2.46–5.38) | 1.10 (0.86–1.29) | | |
| Serology test 3 | 1.12 (0.56–2.07) | 2.52 (1.80–3.99) | 0.67 (0.37–0.93) | | |
| **Gender** | | | | | 0.479 |
| Female | 1675 (66.3) | 186 (11.1%) | 1484 (88.9%) | 11.1% | |
| Male | 852 (33.7) | 87 (10.2%) | 765 (89.8%) | 10.2% | |
| **Nationality** | | | | | 0.028 |
| Non-Saudi | 1654 (65.4) | 195 (11.8%) | 1456 (88.2%) | 11.8% | |
| Saudi | 874 (34.6) | 78 (8.9%) | 794 (91.1%) | 8.9% | |
| **Smoking Status** | | | | | <0.001 |
| Smokers | 355 (14.1%) | 19 (5.4%) | 336 (94.7%) | 5.4% | |
| Non-smokers | 2168 (85.9%) | 254 (11.7%) | 1914 (88.3%) | 11.7% | |
| **Flu Vaccine** | | | | | 0.402 |
| Yes | 791 (31.3%) | 59 (7.5%) | 731 (92.5%) | 7.5% | |
| No | 331 (13.1%) | 20 (6.1%) | 310 (93.9%) | 6.1% | |
| **Positive PCR Test** | | | | | <0.001 |
| Yes | 290 (11.5%) | 208 (72.0%) | 81 (28.0%) | 72.0% | |
| No/ or not done | 2238 (88.5%) | 656 (2.9%) | 2169(94.0%) | 2.9% | |
| **Occupation** | | | | | 0.080 |
| Doctor | 456 (18.0%) | 45 (9.9%) | 410 (90.1%) | 9.9% | |
| Nurse | 1351 (53.0%) | 153 (11.3%) | 1196 (88.7%) | 11.3% | |
| Non-clinical Staff | 277 (11.0%) | 37 (13.4%) | 240 (86.6%) | 13.4% | |
| Infection Control Specialist | 21 (0.8%) | 2 (9.5%) | 19 (90.5%) | 9.5% | |
| Allied Healthcare | 205 (8.1%) | 11 (5.4%) | 192 (90.6%) | 5.4% | |
| Pharmacist | 95 (3.8%) | 9 (9.5%) | 86 (90.5%) | 9.5% | |
| Respiratory Therapist | 40 (1.6%) | 8 (20.0%) | 32 (80.0%) | 20.0% | |
| Others | 83 (3.3%) | 8 (9.6%) | 75 (90.4%) | 9.6% | |
| **Blood Group** | | | | | 0.784 |
| A+ | 614 (24.3%) | 75 (12.3%) | 537 (87.8%) | 12.3% | |
| A- | 37 (1.5%) | 3 (8.1%) | 34 (91.9%) | 8.1% | |
| AB+ | 147 (5.8%) | 15 (10.2%) | 132 (89.8%) | 10.2% | |
| AB- | 12 (0.5%) | 1 (8.3%) | 11 (91.7%) | 8.3% | |
| B+ | 524 (20.7%) | 56 (10.7%) | 466 (89.3%) | 10.7% | |
| B- | 24 (1.0%) | 4 (16.7%) | 20 (83.3%) | 16.7% | |
| O+ | 1031 (40.8%) | 111 (10.8%) | 919 (89.2%) | 10.8% | |
| O- | 77 (3.1%) | 5 (6.5%) | 72 (93.5%) | 6.5% | |
| **Previous medications** | | | | | |
| ACE inhibitors | 92 (3.6%) | 12 (13.0%) | 80 (87.0%) | 13.0% | 0.447 |
| Statins | 113 (4.5%) | 15 (13.3%) | 98 (86.7%) | 13.3% | 0.360 |
| Immuno-modular Agent | 16 (0.6%) | 0 (0.0%) | 16 (100.0%) | 0.0% | |
| Steroids | 49 (2.0%) | 9 (18.4%) | 40 (81.6%) | 18.4% | 0.080 |
| **Medical conditions** | | | | | |
| Cancer | 21 (0.8%) | 0 (0%) | 21 (100%) | 0% | 0.109 |
| Chronic Lung Disease | 51 (2.0%) | 7 (13.7%) | 44 (86.3%) | 13.7% | 0.500 |
| Diabetes | 150 (5.9%) | 24 (16.0%) | 126 (84.0%) | 16.0% | 0.035 |

(*Continued*)

**Table 1.** (Continued)

| Characteristics | All participants, n (%) | Seropositive, n (%) | Seronegative, n (%) | Seroprevalence, (%) | P value |
|---|---|---|---|---|---|
| Hypertension | 244 (9.7%) | 43 (17.6%) | 201 (82.4%) | 17.6% | <0.001 |
| Pregnancy | 42 (1.7%) | 6 (14.3%) | 36 (85.7%) | 14.3% | 0.466 |

*some people's gender info missing.

therapist (20%), non-clinical staff had prevalence rate of 13.4%. Nurses and physicians had seropositive rates of 11.3 and 9.9% respectively.

Of the 273 positive HCWs, 80 (29%) had an IgG value of $\geq 1.4$ to $\leq 3.99$ and 193 (71%) had a value $> 3.99$. Thus, we conclude that the vast majority of positive individuals have moderate-to-high titers of NC antibodies.

Age and sex were not statistically significantly different among staff with or without antibodies (median age, 37 (31–44) vs 36 (31–43) years; 87/273 [32%] vs 765/2249 [34%] male). Table 2. Being a non-smoker, non-Saudi, previously diagnosed with COVID-19 infection by PCR, and having diabetes and hypertension were statistically significantly different by seroprevalence status Table 1.

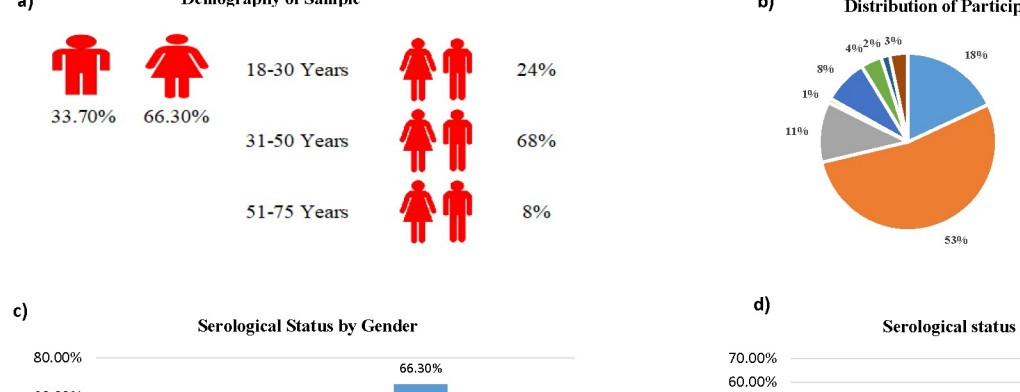

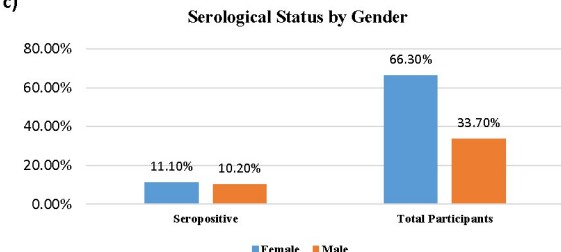

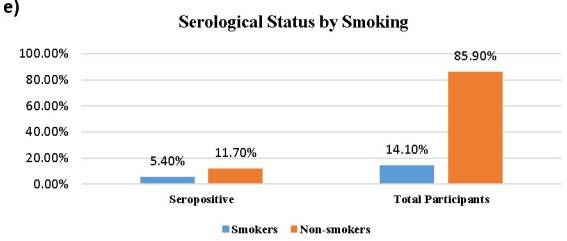

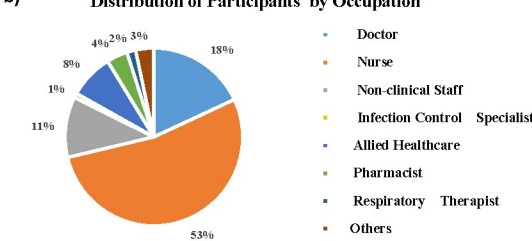

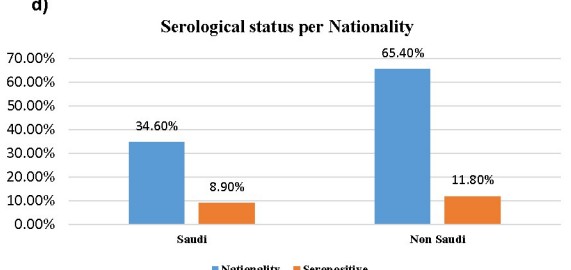

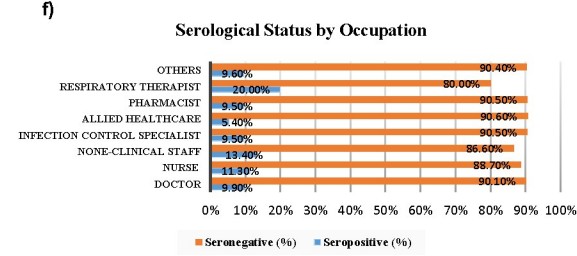

**Fig 1. Sociodemographic and clinical characteristics of the study population.** (A) Demography of sample. (B) Distribution of Participants by Occupation. (C) Serological Status by Gender. (D) Serological status per Nationality. E) Serological Status by Smoking. (F) Serological Status by Occupation.

**Table 2. Demographics and clinical characteristics of study participants by gender.**

| Characteristics | All participants, n (%) | Gender | | P value |
|---|---|---|---|---|
| | | Female | Male | |
| n | 2528 | 1669 (66.2%) | 851 (33.8%) | |
| Age | 36 (31–43) | 36 (31–42) | 37 (31–43) | 0.042 |
| Serology test 1 | 0.03 (0.02–0.07) | 0.03 (0.02–0.07) | 0.03 (0.02–0.07) | 0.355 |
| Serology test 2 | 3.73 (2.22–5.24) | 3.91 (3.32–5.56) | 3.01 (1.85–4.89) | 0.072 |
| Serology test 3 | 1.12 (0.56–2.07) | 1.14 (0.57–2.06) | 1.17 (0.60–2.67) | 0.548 |
| **Nationality** | | | | <0.001 |
| Non-Saudi | 1654 (65.4) | 1265 (76.8%) | 382 (23.19%) | |
| Saudi | 874 (34.6) | 404 (46.3%) | 469 (53.7%) | |
| **Smoking Status** | | | | <0.001 |
| Smokers | 355 (14.1%) | 69 (19.4%) | 286 (80.6%) | |
| Non-smokers | 2168 (85.9%) | 1600 (73.9%) | 565 (26.1%) | |
| **Flu Vaccine** | | | | 0.530 |
| Yes | 791 (31.3%) | 468 (59.3%) | 321 (40.7%) | |
| No | 331 (13.1%) | 203 (61.3%) | 128 (38.7%) | |
| **Positive PCR Test** | | | | 0.554 |
| Yes | 290 (11.5%) | 183 (64.7%) | 100 (35.3%) | |
| No/ or not done | 2238 (88.5%) | 1486 (66.4%) | 751 (33.6%) | |
| **Occupation** | | | | <0.001 |
| Doctor | 456 (18.0%) | 105 (23.0%) | 351 (77.0%) | |
| Nurse | 1351 (53.0%) | 1190 (88.5%) | 155 (11.5%) | |
| Non-clinical Staff | 277 (11.0%) | 156 (56.3%) | 121 (43.7%) | |
| Infection Control Specialist | 21 (0.8%) | 17 (81.0%) | 4 (19.0%) | |
| Allied Healthcare | 205 (8.1%) | 87 (42.7%) | 117 (57.3%) | |
| Pharmacist | 95 (3.8%) | 52 (54.7%) | 43 (45.3%) | |
| Respiratory Therapist | 40 (1.6%) | 10 (25.6%) | 29 (74.4%) | |
| Others | 83 (3.3%) | 52 (62.7%) | 31 (37.3%) | |
| **Blood Group** | | | | 0.118 |
| A+ | 614* (24.3%) | 397 (64.9%) | 215 (35.1%) | |
| A- | 37 (1.5%) | 23 (62.2%) | 14 (37.8%) | |
| AB+ | 147* (5.8%) | 98 (67.1%) | 48 (32.9%) | |
| AB- | 12 (0.5%) | 9 (75.0%) | 3 (25.0%) | |
| B+ | 524* (20.7%) | 370 (70.9%) | 152 (29.1%) | |
| B- | 24* (1.0%) | 14 (60.9%) | 9 (39.1%) | |
| O+ | 1031* (40.8%) | 675 (65.5%) | 355 (34.5%) | |
| O- | 77 (3.1%) | 42 (54.6%) | 35 (45.4%) | |
| **Previous medications** | | | | |
| ACE inhibitors | 92 (3.6%) | 54 (58.7%) | 38 (41.3%) | 0.110 |
| Statins | 113 (4.5%) | 62 (54.9%) | 51 (45.3%) | 0.009 |
| Immuno-modular Agent | 16 (0.6%) | 11 (68.8%) | 5 (31.2%) | 0.864 |
| Steroids | 49 (2.0%) | 38 (77.6%) | 11 (22.4%) | 0.111 |
| **Medical conditions** | | | | |
| Cancer | 21 (0.8%) | 18 (85.7%) | 3 (14.3%) | 0.058 |
| Chronic Lung Disease | 51 (2.0%) | 34 (66.7%) | 17 (33.3%) | 0.947 |
| Diabetes | 150* (5.9%) | 100 (67.1%) | 49 (32.9%) | 0.814 |
| Hypertension | 244 (9.7%) | 172 (71.1%) | 70 (28.9%) | 0.094 |

(*Continued*)

**Table 2.** (Continued)

| Characteristics | All participants, n (%) | Gender | | P value |
|---|---|---|---|---|
| Pregnancy | 42 (1.7%) | 40 (95.2%) | 2 (4.8%) | <0.001 |

*some people's gender info missing.

In this cohort, 456/2523(18%) had at least 1 prior symptom, (3 employees did not answer this question). A high proportion of staff with antibodies had prior symptoms 214/273(78%), The proportion of asymptomatic staff with positive serology was 59/273(22%). Most symptoms were significantly associated with positive serology except sore throat.

Prior anosmia was associated with the presence of antibodies, with an odds ratio of 9.25 (P<0.001), as well as fever and cough. When considering comorbidities, positive serology was significantly associated with a lower prevalence in smokers (OR, 0.48; P = 0.003) and a higher prevalence with hypertension (OR, 1.86; P = 0.009). For occupation, Positive serology was significantly associated with being respiratory therapist (OR, 2.38; P = 0.035), allied health care workers were found protected against SARS-CoV-2 infection (OR, 0.43; P = 0.016) (Tables 3 and 4).

## IgG levels trends over time

In comparing overall IgG levels, we observed a decline from the initial median (25th– 75th percentile) titer of 5.86 (3.57–7.04) to a median (25th– 75th percentile) of 3.91 (2.46–5.38) from the first to the second time point and another drop to a median (25th– 75th percentile) of 2.52 (1.80–3.99) for the last time point (Fig 2).

Of the 185 participants who underwent a second serology test,165 were persistently IgG positive. Finally, 112 HCWs had a third serology test, 44 were still IgG positive (Fig 3).

Among 290 PCR-confirmed SARSCoV-2 HCWs, exposure that led to SARS-CoV-2 infection could have occurred in the community or within the hospital setting (patient, or coworkers) and this study explored their perception between these potential sources of exposure. In general, we found colleagues (20.3%), patients (29.0%), community (10.3%), unknown source (36.6%) and no history of COVID-19 cases contact (3.8%) Table 5.

**Table 3. Risk factors (symptoms) associated with positive SARS-CoV-2 IgG.**

| Symptom | Unadjusted | | Adjusted | |
|---|---|---|---|---|
| | OR (95% CI) | P value | OR (95% CI) | P value |
| Fever | 5.14 (3.43–7.71) | <0.001 | 2.02 (1.16–3.55) | 0.013 |
| Sore throat | 1.05 (0.73–1.52) | 0.779 | 0.58 (0.35–0.95) | 0.032 |
| Vomiting | 6.47 (2.19–19.18) | 0.001 | 3.76 (0.93–15.21) | 0.063 |
| Diarrhoea | 2.97 (1.87–4.72) | <0.001 | 1.19 (0.63–2.25) | 0.584 |
| Chills | 5.55 (3.17–9.73) | <0.001 | 1.73 (0.82–3.65) | 0.151 |
| Muscle ache | 2.95 (2.01–4.33) | <0.001 | 0.96 (0.55–1.70) | 0.894 |
| Cough | 3.81 (2.57–5.63) | <0.11 | 2.09 (1.24–3.51) | 0.005 |
| Loss of smell | 14.91 (9.01–24.69) | <0.001 | 9.25 (5.01–17.05) | <0.001 |
| Fatigue | 3.45 (2.34–5.08) | <0.001 | 1.45 (0.82–2.57) | 0.201 |
| Loss of appetite | 6.37 (3.73–10.88) | <0.001 | 1.14 (0.54–2.38) | 0.733 |
| Nausea | 2.60 (1.55–4.35) | <0.001 | 0.48 (0.22–1.05) | 0.065 |
| Shortness of breath | 2.42 (1.47–3.97) | <0.001 | 0.53 (0.25–1.12) | 0.095 |
| Headache | 2.51 (1.69–3.73) | <0.001 | 1.08 (0.64–1.83) | 0.783 |

**Table 4. Risk factors associated with positive SARS-CoV-2 IgG.**

| Variable | Unadjusted | | Adjusted | |
|---|---|---|---|---|
| | OR | P value | OR (95% CI) | P value |
| Male | 0.91 (0.69–1.19) | 0.479 | - | - |
| Saudi | 0.73 (0.56–0.97) | 0.028 | - | - |
| Smoker | 0.43 (0.36–0.69) | 0.001 | 0.48 (0.29–0.78) | 0.003 |
| Flu vaccine | 1.25 (0.74–2.11) | 0.402 | - | - |
| Occupation | | | | |
| Nurse (reference) | - | - | - | - |
| Doctor | 0.86 (0.60–1.22) | 0.392 | - | - |
| Non-clinical Staff | 1.21 (0.82–1.77) | 0.342 | - | - |
| Infection Control Specialist | 0.82 (0.19–3.57) | 0.616 | - | - |
| Allied Healthcare | 0.45 (0.24–0.84) | 0.013 | 0.43 (0.22–0.85) | 0.016 |
| Pharmacist | 0.82 (0.40–1.66) | 0.578 | - | - |
| Respiratory Therapist | 1.95 (0.88–4.32) | 0.098 | 2.38 (1.10–5.34) | 0.035 |
| Others | 0.83 (0.39–1.76) | 0.634 | - | - |
| Previous medicine | | | | |
| ACE inhibitors | 1.27 (0.68–2.37) | 0.448 | - | - |
| Statins | 1.30 (0.74–2.27) | 0.361 | - | - |
| Immunomodular Agent | - | - | | |
| Steroids | 1.91 (0.91–3.98) | 0.085 | - | - |
| Medical conditions | | | | |
| Cancer | - | - | | |
| CLD | 1.32 (0.59–2.96) | 0.501 | - | - |
| Hypertension | 1.91 (1.33–2.72) | <0.001 | 1.86 (1.17–2.97) | 0.009 |
| Pregnancy | 1.38 (0.58 - .331) | 0.468 | - | - |

## Discussion

Antibody response against the surface S glycoprotein and NC can be detected in most infected individuals 10–15 days after the onset of COVID-19 symptoms [14].

Here, we evaluated the prevalence of SARS-CoV-2 IgG in a wide spectrum of HCWs consisting of frontline, support staff and administrators. We found a seroprevalence of 10.8% among our HCWs.

In a comprehensive review conducted by Galanis *et al.* [15] to estimate the seroprevalence of SARS-CoV-2 antibodies among 12748 HCWs, it was found an overall seroprevalence of 8.7%, ranging from 0% to 45.3% between studies. Seroprevalence in HCWs was higher in studies conducted in North America (12.7%) compared with those conducted in Europe (8.5%), Africa (8.2) and Asia (4%). The following factors were associated with seropositivity: male gender; Hispanic, Asian and Black HCWs; work in a COVID-19 area; front-line HCWs; patient-related work; healthcare assistants; shortage of personal protective equipment; previous positive PCR test; self-reported belief of prior COVID-19 infection and household contact with suspected or confirmed cases of COVID-19 [15].

These discrepancies may be explained by differences in the sensitivity, specificity, performance and design of the assays used, including the antigen targeted as well as differences in the study populations. Seroprevalence might be underestimated if infected individuals had not yet mounted an IgG response or if IgG titers had declined since infection.

In a survey of a recovered patients in the Iceland population, IgG levels were higher in older patients and in those more severely affected by COVID-19. Females tend to become less sick

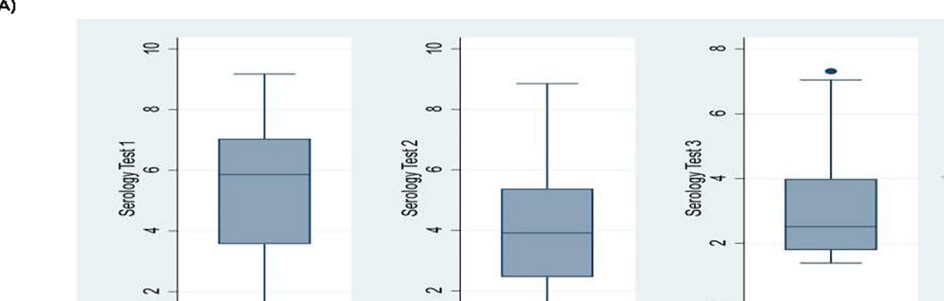

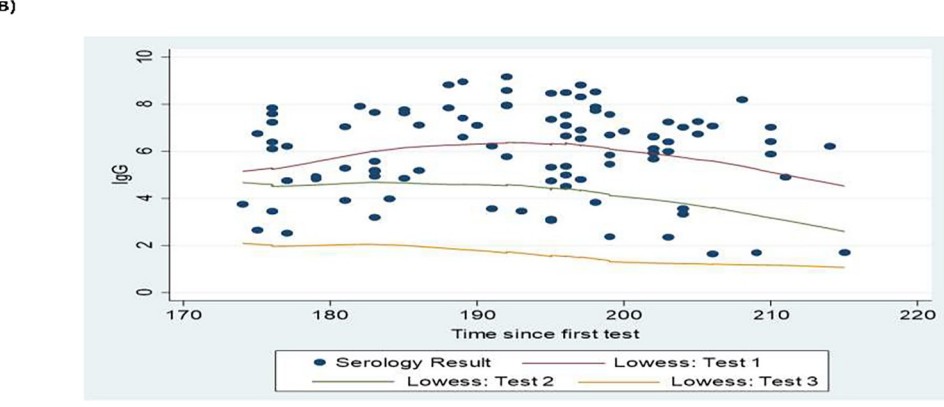

**Fig 2. Trend in the titer of IgG over the study period as.** (A) Median. (B) Line plot (Lowess smoother).

than males and thus had lower IgG levels. IgG levels were lower in smokers, smoking increases the risk of severe Covid-19 illness among young patients [16], and smoking has been observed to increase the expression of ACE2 [17], the receptor for cellular entry of the SARS-CoV-2 virus [7].

Here, we observed that seropositivity was significantly associated with some symptoms with loss of smell having the strongest association with positivity and oddly sore throat was negatively associated with IgG positivity which is similar to a Belgium study and 2 different US studies [18–20]. A study conducted in France found similar association with five clinical symptoms were independently associated with positive serology including asthenia, fever, myalgia, ageusia and anosmia for which the higher odd ratio was observed (11.1 [7.4–16.6; 95% CI]) [21]. We noticed a high risk of infection associated with respiratory therapist job category which they are considered to be a front liner HCWs, opposite to allied health care workers whom we found that they were protected against SARS-CoV-2 infection which could be explained by low rate of direct patients contact. Table 6 outlines some of the IgG longitudinal studies conducted among HCWs [9, 22–26].

The differences observed in SARS-CoV-2 antibody trajectories may be assay and/or antigen dependent, e.g., waning of anti-NC IgG with stable anti-spike IgG using the same Abbott platform as seen in previous studies, but total anti-NC antibodies assayed using a Roche platform remained stable [12, 27]. Probably, these findings depend on assay cut-offs which can be tuned to priorities sensitivity or specificity, with inherently more specific assays having potential to also be set more sensitively, resulting in longer durations of detectable IgG antibodies. For anti-NC, it was observed a higher IgG titers with longer durability occurring after PCR

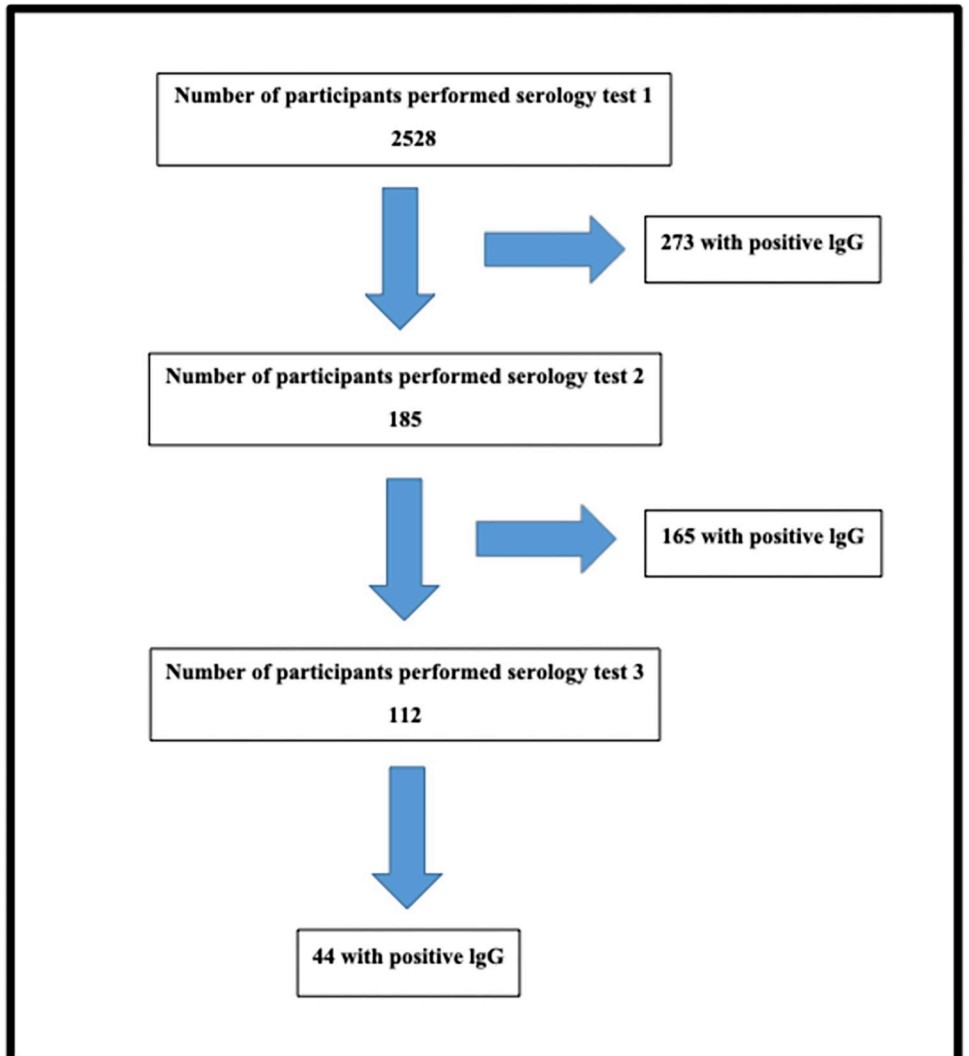

**Fig 3. Flow diagram of serology tests among study participants.** Perceived sources of COVID-19 exposure among study participants.

confirmed infection, consistent with data from Long et al. where 40% of asymptomatic subjects and 13% of the symptomatic individuals became IgG negative in the early convalescent phase [24, 28].

Our results indicate that cross-sectional serosurveys to determine population level immunity may underestimate rates of previous infections. There will be epidemiological implication because if IgG levels fall below detection levels before they are measured, under ascertainment of past infection might occur. Thus, it is critical to understand immune responses to SARS-CoV-2 infection in order to define parameters in which antibody tests can provide meaningful data in the absence of PCR testing in population studies.

## Strength

We included a larger number of subjects compared to most of the previous reported studies. Our study was conducted in multiple clinical sites and mobile teams were utilized, therefore,

**Table 5. Perceived sources of COVID-19 exposure among study participants (n = 290).**

| Sources | Colleagues (n = 59 (20.3%)) | Patients (n = 84 (29.0%)) | Community (n = 30 (10.3%)) | Don't know (n = 106 (36.6%)) | No COVID-19 contact (n = 11 (3.8%)) | P value |
|---|---|---|---|---|---|---|
| **Nationality** | | | | | | 0.068 |
| Non-Saudi | 30 (16.5%) | 62 (34.1%) | 20 (11.0%) | 64 (35.2%) | 6 (3.3%) | |
| Saudi | 29 (26.9%) | 22 (20.4%) | 10 (9.3%) | 42 (38.9%) | 5 (4.6%) | |
| **Occupation** | | | | | | [a] |
| Doctor | 10 (17.9%) | 20 (35.7%) | 3 (5.4%) | 22 (39.3%) | 1 (1.8%) | |
| Nurse | 22 (14.2%) | 55 (35.5%) | 17 (11.0%) | 56 (36.1%) | 5 (3.2%) | |
| Non-clinical Staff | 12 (32.4%) | 5 (13.5%) | 4 (10.8%) | 14 (37.8%) | 2 (5.4%) | |
| Infection Control Specialist | 1 (33.3%) | 0 (0%) | 1 (33.3%) | 1 (33.3%) | 0 (0%) | |
| Allied Healthcare | 6 (33.3%) | 3 (16.7%) | 2 (11.1%) | 5 (27.8%) | 2 (11.1%) | |
| Pharmacist | 4 (50.0%) | 0 (0%) | 1 (12.5%) | 3 (37.5%) | 0 (0%) | |
| Respiratory Therapist | 3 (42.9%) | 1 (14.3%) | 0 (0%) | 2 (28.6%) | 1 (14.3%) | |
| Others | 1 (16.7%) | 0 (0%) | 2 (33.3%) | 3 (50.0%) | 0 (0%) | |

[a] Due to small numbers in many cells over several rows, exact test could not be performed.

**Table 6. Summary of some important longitudinal SARS-CoV-2 IG studies among Health Care Workers.**

| Country | Sample size | Clinical severity of the study population | Assay used With antigen target | Starting point | Duration |
|---|---|---|---|---|---|
| UK[9] | 37 | symptomatic and asymptomatic | S glycoprotein, RBD and N protein were measured by (ELISA) | POS | Decline within 94 days, varying with the initial peak response and diseases severity. |
| USA[23] | 249 | Asymptomatic-mild | A validated enzyme-linked immunosorbent assay against the prefusion-stabilized extracellular domain of the SARS-CoV-2 spike protein. | Baseline Positive serology | 8/19 (42%) persist for 60 days |
| Belgium[24] | 850 | 5 were asymptomatic, 75 had reported mild symptoms, and 1 hospitalized | Antibodies targeting S1 (spike subunit 1) protein with a commercial semi-quantitative enzyme-linked immunosorbent assay (ELISA) (Euroimmun IgG; Medizinische Labordiagnostika, Lübeck, Germany) | PSO[a] or positive PCR[b] for asymptomatic patients(day of first positive serological test minus 14 days). | 74 (91%) who remained seropositive, median duration of antibody persistence |
| | | | | | 168·5 (range 62–199) days. 71 (96%) of 74 HCWs[c] have already had antibodies for 90 days or more and 67 (91%) have had them for 120 days or more |
| UK[25] | 3276 | Asymptomatic and Symptomatic | Anti-trimeric-spike IgG levels were measured using an ELISA developed by the University of Oxford, Abbott Architect i2000 chemiluminescent microparticle immunoassay (CMIA; Abbott, Maidenhead, UK) | Positive serology | Median of 4 months from their maximum IgG titre. |
| USA[26] | 3,458 | Asymptomatic and mild symptoms | Anti-spike IgG antibodies—Ortho Clinical Diagnostics VITROS® XT 7600 platform | 8 weeks after the first blood sample | all of our sero-positive HCWs have maintained antibody positivity for at least 8 weeks, |
| Spain[27] | 578 | Mild (a symptomatic and symptomatic) | Magnetic microspheres from Luminex Corporation (Austin, TX) against receptor-binding domain (RBD) of the spike glycoprotein of SARS-CoV-2 | PSO[a] or positive PCR[b] | • (3.08%) seroconverted for IgG at 3 months follow up.<br>• Decay rate 0.66 (95% CI, 0.53; 0.82) |

[a]PSO: Post symptoms onset.

[b]Polymerase chain reaction.

[c]HCWs: Health Care Workers.

potentially more representative of the overall prevalence of SARS-CoV-2 infectivity amongst HCWs in the workplace with variable exposure to SARS-CoV-2 at their job.

Most published surveys are predominantly cross-sectional or at most include a longitudinal follow-up of short duration with few of them extend up to 6–8 months' Here, a longitudinal data for IgG positive subjects were conducted for prolonged duration to determine the kinetics of SARS-CoV-2 antibodies with at least two time points per subject.

COVID-19 infections are predominantly mild or even asymptomatic. While the immunological responses to severe COVID-19 are relatively well described [29, 30] understanding the response in mild COVID-19 cases is required, since mild and asymptomatic cases constitute the majority of our cohort. It was crucial to understand the robustness of the antibody response in mild cases, including its longevity, so as to inform serosurveys, as well as to determine levels and duration of antibody titers.

We used a high-quality serological testing, in a head-to-head comparison of 12 different serology assays for detection of SARS-CoV-2 antibodies, our assay used here found to be among the tests with a highest clinical sensitivity and specificity [13]. Generally, majority of antibodies are directed against the most abundant protein, which is the NC. Therefore, tests that measure IgG to NC would be the most sensitive. However, the receptor-binding domain of S (RBD-S) protein is the host attachment protein, and antibodies against RBD-S are expected to be neutralizing and would be more specific. Therefore, using both antigens for detecting IgG would result in high sensitivity [31].

## Limitation

The current study has some limitations. It is single-center design and testing of only 33% of HCWs, explained by the fact that at least one-third of those not tested were individuals not at work during the study period. Seroconversion may have been missed if testing was too early, especially without IgM results that might reflect more recent infection than IgG.

Antibody responses were only analyzed using 1 antigen and other viral proteins may elicit different responses in different individuals [14], thus we could have slightly underestimated the overall seroprevalence of infection. Previous published studies indicated that NC-specific antibodies waned more quickly than did S-specific antibodies.

It estimates point prevalence of SARS-CoV-2 IgG in HCWs and was not designed to be conducted as periodic serosurveys, which allows monitoring of seroprevalence progression over the epidemic course. Prevalence among HCWs will be dynamic, and likely to be affected as the infection rate across the community changes. This snapshot study was not intended to capture such trends.

Selection bias might happen since the participants may have been more inclined to volunteer if they were concerned about COVID-19 infection. Indeed, it is conceivable that individuals who experienced COVID-19-like symptoms, or those that were less confined during lockdown were more likely to participate in the study, potentially leading to overestimation of our prevalence. HCWs with exposures or symptoms may have been less inclined to report these accurately (information bias), though reassurance about confidentiality will have at least in part mitigated this. The potential for exposure and symptoms recall bias about was present throughout. To reduce the effect of recall bias, all surveys were filled out by HCWs before receiving their serology results.

## Conclusion

This survey found variation in the SARS-CoV-2 seroprevalence in different groups of HCWs. We identified increased risk of infection in frontline staff mainly respiratory therapist which

could be explained by the nature of extensive direct patient contact, the lack of available personal protective equipment early on in the pandemic and participation in aerosolizing procedures which confer significant effect on seropositivity.

This strongly supports the notion that differential risk of SARS-CoV-2 exposure exists within the hospitals. Our findings raise concern that humoral immunity may not be long lasting in patients with mild illness, who represent the majority of Covid-19 patients. It is important to note that the loss of IgG positivity is not equivalent to loss of immunity. However, longitudinal reports are not in full agreement about the longevity of antibody titers, with some showing that IgG levels are waning rapidly by approximately several weeks after infection while others reporting stable levels detected over months, and whether protective immunity will be maintained with a lower antibody titer is unknown.

There are important questions that need to be answered with appropriately designed studies. Importantly, we need to define the specific antibody titers that correlate of protection. A combination of a detailed knowledge of specific antibody dynamic plus determining protective titers would help us to make predictions who is at a reduced risk of reinfection.

## Acknowledgments

Special thanks to the HCWs who participated in this study and all HCW who put their own lives at risk during the pandemic.

## Author Contributions

**Conceptualization:** Faisal Alasmari, Mahmoud Mukahal, Alaa Ashraf Alqurashi.

**Data curation:** Fatima Alabdrabalnabi, Abdullah AlJurayyan, Shymaa Moshobab Alkahtani, Fatimah Salem Assari, Rahaf Bashaweeh, Rana Salam, Solaf Aldera, Ohud Mohammed Alkinani, Kholoud AlEnizi.

**Formal analysis:** Molla Huq.

**Methodology:** Molla Huq, Talal Almutairi.

**Resources:** Solaf Aldera.

**Validation:** Abdullah AlJurayyan, Talal Almutairi.

**Writing – original draft:** Faisal Alasmari.

**Writing – review & editing:** Mahmoud Mukahal, Alaa Ashraf Alqurashi, Imad Tleyjeh.

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
