## [Decision Letter · Decision Letter 0]

1 Mar 2022

PONE-D-21-40113Seroprevalence and longevity of SARS-CoV-2 nucleocapsid antigen-IgG among health care  workers in a large COVID-19 public hospital in Saudi Arabia: A prospective cohort studyPLOS ONE

Dear Dr. Alasmari,

Thank you for submitting your manuscript to PLOS ONE. After careful consideration, we feel that it has merit but does not fully meet PLOS ONE’s publication criteria as it currently stands. Therefore, we invite you to submit a revised version of the manuscript that addresses the points raised during the review process.

We look forward to receiving your revised manuscript.

Kind regards,

Asep K. Supriatna, Ph.D

Academic Editor

PLOS ONE

Journal Requirements:

5. Thank you for submitting the above manuscript to PLOS ONE. During our internal evaluation of the manuscript, we found significant text overlap between your submission and the following previously published works, some of which you are an author.

- https://www.medrxiv.org/content/10.1101/2020.07.09.20150136v1

- https://pubmed.ncbi.nlm.nih.gov/33341124/

- https://discovery.ucl.ac.uk/id/eprint/10117965/2/Walker_The%20duration%2C%20dynamics%20and%20determinants%20of%20SARS-CoV-2%20antibody%20responses%20in%20individual%20healthcare%20workers_AAM.pdf

Please revise the manuscript to rephrase the duplicated text, cite your sources, and provide details as to how the current manuscript advances on previous work. Please note that further consideration is dependent on the submission of a manuscript that addresses these concerns about the overlap in text with published work.

Additional Editor Comments:

Dear Authors,

We have reached a decision regarding the submitted manuscripts. Please revise your manuscript by adhering to the comment of the reviewers (and also to accomodate the reviewers' suggestion) as attached.

Best regards,

Academic Editor

Reviewers' comments:

Reviewer's Responses to Questions

**Comments to the Author**

1. Is the manuscript technically sound, and do the data support the conclusions?

Reviewer #1: Partly

Reviewer #2: Partly

2. Has the statistical analysis been performed appropriately and rigorously? 

Reviewer #1: No

Reviewer #2: No

3. Have the authors made all data underlying the findings in their manuscript fully available?

Reviewer #1: Yes

Reviewer #2: Yes

4. Is the manuscript presented in an intelligible fashion and written in standard English?

Reviewer #1: No

Reviewer #2: Yes

5. Review Comments to the Author

Reviewer #1: My appreciation goes for the article. Here are some points to note:

1.For writing techniques, please refer to the publisher formats and articles that have been published. For example: the abstract should also contain the methods, the essential results, and conclusions in one complete paragraph; the second paragraph in line 93-95 one paragraph consists only 1 sentence.

2.In the abstract:

(i)Important results from statistical surveys, and results from multivariable logistic regression should be added.

(ii)The main results of the demographic statistics (Table 1) that specifically consider three factors: IgG seropositive status, clinical, and epidemiological respectively, should also be included.

3.In the methods:

(i)HCW should be associated with the location where they work. It must be mapped what are four hospitals in one city look like, such as the capacity, etc. (related with line 117)

(ii)Statictics methods should be explain more clearly particularly in multivariate logistic regression. Note: Is the multivariable logistic regression same with the multivariate logistic regression?

(iii)Alpha used is too large. It should be 0.01.

4. In the results:

(i)Please show the multivariable (multiple) logistic regression model, follows by the estimate parameters, chi-square table, etc. The example of mathematical model is as follows:

(ii)The descriptive statistics is more attractive and interesting if it is represented by figure/graph, for example:

B: Proportion of SARAS-Cov-2 samples across cities and regencies in West Java, Indonesia.

C: Distribution of SARS-CoV-2 sequenced samples across different cities and regencies in West Java Indonesia.

D: Demography of samples.

Source: Azzania, et al. (2021). Analysis of SARS-CoV-2 Genomes from West Java, Indonesia. Viruses (13), 2097.

(iii)Comparisons should suffice in unrelenting sentences. (What is the comparison table for?)

(iv)Demographics should be linked to where they live.

(v)Add an analysis of the effects of gender on that demographic, i. e. the effect of these results (3 factors: IgG seropositive status, clinical, and epidemiological) on each sex.

Reviewer #2: 1. The objective of the study should be clearly stated in the abstract.

2. Table 3 (line 186): All the p-values are for the test of significance of the variables not for the regression coefficients. For e.g., there should be a p-value for "occupation", not for certain types of occupation (allied, resp. therapist) - consult your statistician.

3. Line 193: Fig 1. Trend in mean titer of IgG over the study period. A boxplot is not plotting a mean but a median (Q1, Q2=median, Q3). It would be informative if the titer of IgG are visualized as a line plot per individuals with time and provided by an average line plot for all titer of IgG. A lowess smoothing plot can be used for the average line plot. Since the individuals are a lot, a soft tone color (grey color, for example) for all individual plot is used. The color for smoothing plot should be bold and strong (black color, for example). An appropriate time scale should be used for the plot, if the date of the serologic tests were available, use "the time since the PCR test" or "the time since the first serologic test" (Choose the best available starting point). Consult your statistician to do such plot.

4. There is no sound Conclusion (Line 309). There is no answer (at least a comment) to how occupation affect seropositive outcomes (the risk factors), for example. How long the duration of detectable IgG, and how it is as compare to other countries.

5. Table 1 (Line 163), Characteristics n: there were 5 subjects (2528-273-22650=5) unclassified (seropositive or seronegative). This should be reported either in the table or in the text.

6. Be consistent with using or not using a thousand separator (,), for e.g. in line 118: "... a total of 7737 HCWs: 1,021 medical staff.

6. PLOS authors have the option to publish the peer review history of their article (what does this mean?). If published, this will include your full peer review and any attached files.

Reviewer #1: No

Reviewer #2: No

---

## [Author Response · Author response to Decision Letter 0]

15 Apr 2022

Academic editor Response 

1. Please ensure that your manuscript meets PLOS ONE's style requirements, including those for file naming. 1. DONE

In your Data Availability statement, you have not specified where the minimal data set underlying the results described in your manuscript can be found. PLOS defines a study's minimal data set as the underlying data used to reach the conclusions drawn in the manuscript and any additional data required to replicate the reported study findings in their entirety. All PLOS journals require that the minimal data set be made fully available. For more information about our data policy, please see http://journals.plos.org/plosone/s/data-availability.

We note that you have stated that you will provide repository information for your data at acceptance. Should your manuscript be accepted for publication, we will hold it until you provide the relevant accession numbers or DOIs necessary to access your data. If you wish to make changes to your Data Availability statement, please describe these changes in your cover letter and we will update your Data Availability statement to reflect the information you provide. 

4. DONE

5. Thank you for submitting the above manuscript to PLOS ONE. During our internal evaluation of the manuscript, we found significant text overlap between your submission and the following previously published works, some of which you are an author.

- https://www.medrxiv.org/content/10.1101/2020.07.09.20150136v1-
https://pubmed.ncbi.nlm.nih.gov/33341124/https://discovery.ucl.ac.uk/id/eprint/10117965/2/Walker_The%20duration%2C%20dynamics%20and%20determinants%20of%20SARS-CoV-2%20antibody%20responses%20in%20individual%20healthcare%20workers_AAM.pdf

Please revise the manuscript to rephrase the duplicated text, cite your sources, and provide details as to how the current manuscript advances on previous work. Please note that further consideration is dependent on the submission of a manuscript that addresses these concerns about the overlap in text with published work.

We will carefully review your manuscript upon resubmission, so please ensure that your revision is thorough. 5. DONE

Reviewer #1 

My appreciation goes for the article. Here are some points to note:

1.For writing techniques, please refer to the publisher formats and articles that have been published. For example: the abstract should also contain the methods, the essential results, and conclusions in one complete paragraph; the second paragraph in line 93-95 one paragraph consists only 1 sentence. 1. DONE

2.In the abstract:

(i)Important results from statistical surveys, and results from multivariable logistic regression should be added.

(ii)The main results of the demographic statistics (Table 1) that specifically consider three factors: IgG seropositive status, clinical, and epidemiological respectively, should also be included. 2. (i) DONE

(ii) DONE

3.In the methods:

(i)HCW should be associated with the location where they work. It must be mapped what are four hospitals in one city look like, such as the capacity, etc. (related with line 117) 3. (i)Thank you for your comments, this information is not available. Line 117 has been modified 

(ii)Statistics methods should be explain more clearly particularly in multivariate logistic regression. Note: Is the multivariable logistic regression same with the multivariate logistic regression?

(iii)Alpha used is too large. It should be 0.01. (ii)We did NOT use multivariate logistic regression, we used multivariable/multiple logistic regression. Logistic regression formula.

(iii)We applied the standard practice of using alpha 0.05.

4. In the results:

(i)Please show the multivariable (multiple) logistic regression model, follows by the estimate parameters, chi-square table, etc. The example of mathematical model is as follows:

(ii)The descriptive statistics is more attractive and interesting if it is represented by figure/graph, for example:

B: Proportion of SARAS-Cov-2 samples across cities and regencies in West Java, Indonesia.

C: Distribution of SARS-CoV-2 sequenced samples across different cities and regencies in West Java Indonesia.

D: Demography of samples.

Source: Azzania, et al. (2021). Analysis of SARS-CoV-2 Genomes from West Java, Indonesia. Viruses (13), 2097. (i) Below is the model used in the study and we did not see the need to add it in the manuscript. 

Univariate logistic regression: The probability function is as below:

p=ⅇ^(β_0+β_1 x)/(1+ⅇ^(β_0+β_1 x) )

Multivariable or Multiple logistic regression: The probability function for multiple explanatory variables is as below:

p=ⅇ^(β_0+β_1 x_1+β_2 x_2+β_3 x_3+⋯.β_n x_n )/(1+e^(β_0+β_1 x_1+β_2 x_2+β_3 x_3+⋯.β_n x_n ) )

(ii) DONE, we added figures of descriptive statistics.

(iii)Comparisons should suffice in unrelenting sentences. (What is the comparison table for?)

(iv)Demographics should be linked to where they live. (iii) Thanks for this important comment, the written sentences are enough and the data in the table are more elaborative.

(iv) Thank you for your comment. We don't have location data in our dataset.

(v)Add an analysis of the effects of gender on that demographic, i. e. the effect of these results (3 factors: IgG seropositive status, clinical, and epidemiological) on each sex. (v) Table added

Reviewer #2: 

1.The objective of the study should be clearly stated in the abstract. 1. The objective has been added.

2. Table 3 (line 186): All the p-values are for the test of significance of the variables not for the regression coefficients. For e.g., there should be a p-value for "occupation", not for certain types of occupation (allied, resp. therapist) - consult your statistician. 2. We have used occupation in a cross tabulation (chi-square test) in Table 1 but in univariable regression analysis we used dummy variables for each occupational type. Hence, they are statistically sound.

3. Line 193: Fig 1. Trend in mean titer of IgG over the study period. A boxplot is not plotting a mean but a median (Q1, Q2=median, Q3). It would be informative if the titer of IgG are visualized as a line plot per individuals with time and provided by an average line plot for all titer of IgG. 

A lowess smoothing plot can be used for the average line plot. Since the individuals are a lot, a soft tone color (grey color, for example) for all individual plot is used. The color for smoothing plot should be bold and strong (black color, for example). An appropriate time scale should be used for the plot, if the date of the serologic tests were available, use "the time since the PCR test" or "the time since the first serologic test" (Choose the best available starting point). Consult your statistician to do such plot. 3. DONE (Figures added)

4. There is no sound Conclusion (Line 309). There is no answer (at least a comment) to how occupation affect seropositive outcomes (the risk factors), for example. How long the duration of detectable IgG, and how it is as compare to other countries. 4. Conclusion has been modified and table 6 in discussion demonstrated the differences among countries. 

5. Table 1 (Line 163), Characteristics n: there were 5 subjects (2528-273-22650=5) unclassified (seropositive or seronegative). This should be reported either in the table or in the text. 5. DONE. Stated in the manuscript in abstract and result 

6. Be consistent with using or not using a thousand separator (,), for e.g. in line 118: "... a total of 7737 HCWs: 1,021 medical staff.

 6. DONE

---

## [Decision Letter · Decision Letter 1]

12 Jun 2022

PONE-D-21-40113R1Seroprevalence and longevity of SARS-CoV-2 nucleocapsid antigen-IgG among health care  workers in a large COVID-19 public hospital in Saudi Arabia: A prospective cohort studyPLOS ONE

Dear Dr. Alasmari,

Thank you for submitting your manuscript to PLOS ONE. After careful consideration, we feel that it has merit but does not fully meet PLOS ONE’s publication criteria as it currently stands. Therefore, we invite you to submit a revised version of the manuscript that addresses the points raised during the review process.

We look forward to receiving your revised manuscript.

Kind regards,

Asep K. Supriatna, Ph.D

Academic Editor

PLOS ONE

Additional Editor Comments:

Dear authors,

Thank you for submiting the revised version of the manuscript. The revised manuscript has been evaluated by the reviewer and there are still some issues need to be addressed. Please make correction based on the comment of the reviewer.

Best regards,

Reviewers' comments:

Reviewer's Responses to Questions

**Comments to the Author**

1. If the authors have adequately addressed your comments raised in a previous round of review and you feel that this manuscript is now acceptable for publication, you may indicate that here to bypass the “Comments to the Author” section, enter your conflict of interest statement in the “Confidential to Editor” section, and submit your "Accept" recommendation.

Reviewer #2: (No Response)

2. Is the manuscript technically sound, and do the data support the conclusions?

Reviewer #2: Partly

3. Has the statistical analysis been performed appropriately and rigorously? 

Reviewer #2: No

4. Have the authors made all data underlying the findings in their manuscript fully available?

Reviewer #2: No

5. Is the manuscript presented in an intelligible fashion and written in standard English?

Reviewer #2: Yes

6. Review Comments to the Author

Reviewer #2: Table 4 is still wrong (previously it was Table 3).

All the p-values should be significance test for the variables notfor the regression coefficients. For e.g., there should be a p-value for 'occupation., not for certain types of occupation ('allied', 'resp. therapist', etc.).

The p-value is the simultaneous test for the all dummy variables created by 'occupation' variable. Using this p-value you can evaluate the significance of occupation to the outcome (similar to that of the chi-square test but it is now adjusted for other variables). This is a typical mistake in using regression with dummy variables (with more than 2 categories). Do the same way for all variables with more than 2 categories ('previous medicine', 'medical condition', ... why not include the 'blood group' ...as in the chi-square test?). A sound conclusion then can be made from these p-values: what is the effect of "occupation", what is the effect of "previous medicine", etc.

I suggest you also performing variable selection (model building). In a software like R, there is available procedure like dropterm() to calculate the simultaneous test, stepAIC() for variable selection. It should be available also in STATA, perhaps the TEST syntax.

Fig 2.C

The horizzontal axis should be the time, and the vertical axis is the IgG results. If the date of the test available, it would be informative if the time is "the time since the first serologic test" (not just Test 1, Test 2, Test 3). The attached figure is an example of such a plot. The individual plots are for all 185 (?) participants.

The number of days since the first test may not be the same for all participants, even if the measurement time is planned every 3 months, therefore the horizontal axis is not just Test 1, Test 2, Test 3. (in the example, I made day 0 is the baseline and the Test 1 is participants who are underwent the next test (In your data, the baseline is the Test 1).

Fig 2.B and a plot as suggested in the attached file are informative enough.

7. PLOS authors have the option to publish the peer review history of their article (what does this mean?). If published, this will include your full peer review and any attached files.

Reviewer #2: No

---

## [Author Response · Author response to Decision Letter 1]

17 Jun 2022

To: Asep K. Supriatna, Ph.D

Academic Editor

PLOS ONE

Re: PONE-D-21-40113R1

Title: Seroprevalence and longevity of SARS-CoV-2 nucleocapsid antigen-IgG among health care workers in a large COVID-19 public hospital in Saudi Arabia: A prospective cohort study

We would like to thank the Editorial Office and the reviewers for their time and valuable comments. We appreciate your efforts in critically reviewing our manuscript and we believe these suggestions would add more depth to our manuscript. We hope we have addressed your concerns appropriately. We look forward to seeing our manuscript published in PLOS ONE. Please find our response to your queries below:

Academic Editor 

4. Have the authors made all data underlying the findings in their manuscript fully available? Yes. Submitted in the link

Reviewer #2: 

1. Table 4 is still wrong (previously it was Table 3).

All the p-values should be significance test for the variables not for the regression coefficients. For e.g., there should be a p-value for 'occupation., not for certain types of occupation ('allied', 'resp. therapist', etc.).

The p-value is the simultaneous test for the all dummy variables created by 'occupation' variable. Using this p-value you can evaluate the significance of occupation to the outcome (similar to that of the chi-square test but it is now adjusted for other variables). This is a typical mistake in using regression with dummy variables (with more than 2 categories). Do the same way for all variables with more than 2 categories ('previous medicine', 'medical condition', ... why not include the 'blood group' ...as in the chi-square test?). A sound conclusion then can be made from these p-values: what is the effect of "occupation", what is the effect of "previous medicine", etc.

I suggest you also performing variable selection (model building). In a software like R, there is available procedure like dropterm() to calculate the simultaneous test, stepAIC() for variable selection. It should be available also in STATA, perhaps the TEST syntax. 1. Thanks for your comment. We have assessed the association between occupation as a whole and serology test (outcome) in Table 1 using chi-square test and found it had a trend to significance (p = 0.08). That's why we tried to test using logistic regression to pinpoint whether any particular occupation is related to serology positivity compared to others. Unfortunately, in logistic regression we can't treat occupation as one single variable/category as it has 8 categories. If we do then Stata will treat it as a continuous variable. Hence, we must declare that it's a categorical variable and must declare which category will be the reference category (usually the category which holds most people (considered to be "normal"). That's why in Stata, we declared occupation to be a categorical variable and declared "Nurse" to be the :reference category" as this category held the highest number of people (n = 1351 (53.0%)) of all participants. Hence, we got OR and p value for all the other occupations compared to Nurses. We have also done model building by taking only those variables from simple logistic regression (unadjusted) that had p values less than 0.3 (either significant or potential to become significant). That's why you will see only a few variables in the multiple logistic regression (adjusted) section. As per your suggestion, we have done model building with variable selection. Please see new table-4 in the manuscript.

2. Fig 2.C

The horizontal axis should be the time, and the vertical axis is the IgG results. If the date of the test available, it would be informative if the time is "the time since the first serologic test" (not just Test 1, Test 2, Test 3). The attached figure is an example of such a plot. The individual plots are for all 185 (?) participants.

The number of days since the first test may not be the same for all participants, even if the measurement time is planned every 3 months, therefore the horizontal axis is not just Test 1, Test 2, Test 3. (in the example, I made day 0 is the baseline and the Test 1 is participants who are underwent the next test (In your data, the baseline is the Test 1).

Fig 2.B and a plot as suggested in the attached file are informative enough. We thank the reviewer for his valuable suggestion. We have created a new lowess smoothing graph (figure 2-B).

---

## [Editor Report · Decision Letter 2]

12 Jul 2022

PONE-D-21-40113R2Seroprevalence and longevity of SARS-CoV-2 nucleocapsid antigen-IgG among health care  workers in a large COVID-19 public hospital in Saudi Arabia: A prospective cohort studyPLOS ONE

Dear Dr. Alasmari,

Thank you for submitting your manuscript to PLOS ONE. After careful consideration, we feel that it has merit but does not fully meet PLOS ONE’s publication criteria as it currently stands. Therefore, we invite you to submit a revised version of the manuscript that addresses the points raised during the review process.

We look forward to receiving your revised manuscript.

Kind regards,

Asep K. Supriatna, Ph.D

Academic Editor

PLOS ONE

Journal Requirements:

Additional Editor Comments:

Dear Editors,

Thank you for submitting the revised version of the manuscript. I have read the revised manuscript and also the comments from the reviewer. There still some issues that still need to be addressed by following the reviewer's comment before the manuscript is ready for publication.

Best regards, Asep
---

## [Author Response · Author response to Decision Letter 2]

22 Jul 2022

To: Asep K. Supriatna, Ph.D

Academic Editor

PLOS ONE

Re: PONE-D-21-40113R2

Title: Seroprevalence and longevity of SARS-CoV-2 nucleocapsid antigen-IgG among health care workers in a large COVID-19 public hospital in Saudi Arabia: A prospective cohort study

We would like to thank all the editors and reviewers for the valuable comments. We hope we have addressed all the comments appropriately. We look forward to seeing our manuscript published in PLOS ONE. Please find our response to the journal requirement:

Journal Requirement 

 1. Thanks for your comment. We have revised the reference list and completed it according to the journal requirement.

---

## [Editor Report · Decision Letter 3]

27 Jul 2022

Seroprevalence and longevity of SARS-CoV-2 nucleocapsid antigen-IgG among health care workers in a large COVID-19 public hospital in Saudi Arabia: A prospective cohort study

PONE-D-21-40113R3

Dear Dr. Alasmari,

We’re pleased to inform you that your manuscript has been judged scientifically suitable for publication and will be formally accepted for publication once it meets all outstanding technical requirements.

Kind regards,

Asep K. Supriatna, Ph.D

Academic Editor

PLOS ONE
---

## [Editor Report · Acceptance letter]

2 Aug 2022

PONE-D-21-40113R3 

*Seroprevalence and longevity of SARS-CoV-2 nucleocapsid antigen-IgG among health care workers in a large COVID-19 public hospital in Saudi Arabia: A prospective cohort study*

Dear Dr. Alasmari:

I'm pleased to inform you that your manuscript has been deemed suitable for publication in PLOS ONE. Congratulations! Your manuscript is now with our production department. 

Kind regards, 

on behalf of

Dr. Asep K. Supriatna 

Academic Editor

PLOS ONE